# Therapeutic Drug Monitoring of Vedolizumab in Inflammatory Bowel Disease Patients during Maintenance Treatment—TUMMY Study

**DOI:** 10.3390/pharmaceutics15030972

**Published:** 2023-03-17

**Authors:** Merve Sivridaş, Rob H. Creemers, Dennis R. Wong, Paul J. Boekema, Tessa E. H. Römkens, Lennard P. L. Gilissen, Adriaan A. van Bodegraven, Floris C. Loeff, Theo Rispens, Luc J. J. Derijks

**Affiliations:** 1Department of Clinical Pharmacy, Máxima Medical Center, 5504 DB Veldhoven, The Netherlands; 2Department of Gastroenterology, Geriatrics, Internal, and Intensive Care Medicine (COMIK), Zuyderland Medical Center, 6130 MB Sittard-Geleen, The Netherlands; 3Department of Clinical Pharmacy, Pharmacology and Toxicology, Zuyderland Medical Center, 6162 BG Sittard-Geleen, The Netherlands; 4Department of Gastroenterology, Máxima Medical Center, 5504 DB Veldhoven, The Netherlands; 5Department of Gastroenterology, Jeroen Bosch Hospital, 5223 GZ Den Bosch, The Netherlands; 6Department of Gastroenterology and Hepatology, Catharina Hospital Eindhoven, 5623 EJ Eindhoven, The Netherlands; 7Department of Immunopathology, Sanquin Research and Landsteiner Laboratory, Academic Medical Center, University of Amsterdam, 1105 AZ Amsterdam, The Netherlands; 8Department of Clinical Pharmacy and Toxicology, Maastricht University Medical Center, 6229 HX Maastricht, The Netherlands

**Keywords:** vedolizumab, trough level, therapeutic drug monitoring, maintenance, IBD, ulcerative colitis, Crohn’s disease

## Abstract

There are limited data on therapeutic drug monitoring (TDM) in inflammatory bowel disease (IBD) patients treated with vedolizumab (VDZ). Although an exposure–response relation has been demonstrated in the post-induction phase, this relationship is more uncertain in the maintenance phase of treatment. The aim of our study was to determine whether there is an association between VDZ trough concentration and clinical and biochemical remission in the maintenance phase. A prospective, observational multicenter study has been performed on patients with IBD on VDZ in the maintenance treatment (≥14 weeks). Patient demographics, biomarkers, and VDZ serum trough concentrations were collected. Clinical disease activity was scored by the Harvey Bradshaw Index (HBI) for Crohn’s disease (CD) and the Simple Clinical Colitis Activity Index (SCCAI) for ulcerative colitis (UC). Clinical remission was determined as HBI < 5 and SCCAI < 3. Biochemical remission was defined as fecal calprotectin <250 mg/kg and serum CRP <5 mg/L. A total of 159 patients (59 CD, 100 UC) were included. In none of the patient groups, a statistically significant correlation between trough VDZ concentration and clinical remission was observed. Patients in biochemical remission had higher VDZ trough concentrations (*p* = 0.019). In this population, higher trough VDZ concentrations were associated with biochemical remission but not with clinical remission.

## 1. Introduction

Vedolizumab (VDZ) (Entyvio^®^) is a recombinant humanized monoclonal antibody that specifically binds to α4β7 integrin, which is preferentially expressed on gut-homing T helper lymphocytes. This binding inhibits the adhesion of these cells to mucosal-adressin-cell-adhesion molecule-1 (MAdCAM-1), which is mainly found on the endothelial cells of the gut with an essential role in the migration of T lymphocytes into the tissues of the gastrointestinal tract. As a result, it has been ascribed to being more selectively immunosuppressive, primarily affecting the intestinal mucosa [1,2].

Vedolizumab is indicated in patients with inflammatory bowel disease (IBD) who fail conventional therapy with immunosuppressants (including thiopurines and/or methotrexate) or anti-TNF agents. In clinical practice, VDZ currently is most frequently initiated following anti-TNF failure, but this is debated. Some advocate earlier use in a selected group of anti-TNF-naive patients, in which systemic immune suppression may potentially be detrimental (elderly, high risk of infection, malignancies) [3].

Vedolizumab is administered in a fixed dose of 300 mg infusion. In patients with IBD, the volume of distribution of VDZ at steady state is 4.84 L, which is comparable to other therapeutic monoclonal antibodies and roughly equivalent to the volume of plasma. In a population pharmacokinetic investigation, a two-compartment model with zero-order input and parallel linear and nonlinear elimination fits best with the most accurate description of VDZ pharmacokinetics. At low doses, it undergoes a fast, saturable, target-mediated elimination process, and at higher concentrations, it goes through a slower, linear, nonspecific elimination process. Vedolizumab is primarily eliminated through the linear route at therapeutic doses, leading to a t ½ of 25.5 days [4,5].

Exposure–response relationships for drugs can exist when a higher drug concentration results in more (pharmacodynamics) drug effect or when a higher disease activity results in higher drug clearance and thus a lower drug concentration (e.g., in infliximab) [4]. As an exposure–response relationship is probably present for (some) anti-TNF agents, therapeutic drug monitoring (TDM) is already frequently used in anti-TNF therapy, especially for infliximab. This particularly implies monitoring anti-TNF trough concentrations and anti-TNF antibodies to optimize therapy [4,6]. It is suggested in several studies that there is also an exposure–response relationship for VDZ [7,8,9,10,11,12,13]. Additionally, in some of these studies it was suggested that during the early post-induction phase, patients in remission had significantly higher rates of remission above a therapeutic VDZ concentration threshold of 14 mg/L [9,14]. Therefore, the TDM of VDZ has the potential to individualize and optimize the treatment of IBD. In clinical practice, however, VDZ trough concentrations are not yet routinely monitored. In the treatment of IBD, relatively few options are available. For this reason, it is important to optimize therapy whenever possible before considering a switch to another therapy (class). The role of TDM in VDZ therapy during the maintenance phase remains debatable.

The primary aim of this study was to evaluate the association between VDZ trough concentration and clinical and biochemical disease remission in a population of patients with IBD receiving maintenance VDZ treatment. The secondary aim was to evaluate the association between VDZ trough concentration and disease activity. The tertiary aim was to evaluate if patients with an assumed therapeutic VDZ concentration (14 mg/L) had higher remission rates.

## 2. Materials and Methods

### 2.1. Patient Population and Study Design

In this observational cross-sectional multicenter study, we prospectively collected VDZ trough concentrations in patients with IBD in four collaborating teaching hospitals (Máxima Medical Center (Veldhoven), Catharina Hospital (Eindhoven), Jeroen Bosch Hospital (‘s Hertogenbosch) and Zuyderland Medical Center (Sittard-Geleen), The Netherlands) between May 2021 and May 2022.

Patients were eligible for inclusion if they were ≥18 years of age, diagnosed with ulcerative colitis or Crohn’s disease (based on clinical, radiological endoscopic, and histological criteria by the treating physician), and were treated with VDZ (300 mg) for at least 14 weeks after a regular induction scheme at weeks 0, 2, and 6 (optional week 10).

The use of concomitant immunosuppressants (such as thiopurines, ciclosporin, and methotrexate) was allowed. Patient demographics, co-medication, disease duration, VDZ dose (mg), and dosing interval (weeks) were obtained from the electronic patient records.

Blood samples were drawn just before the administration of a regular VDZ infusion (as Ctrough) at the outpatient clinic on the day of treatment. About 5 mL of extra blood was taken with a routine blood sample to determine VDZ trough concentrations. Patient stools were collected (±2 weeks before or after VDZ administration) for the determination of fecal calprotectin. In the abovementioned blood sample, albumin, hemoglobin, and CRP were routinely determined in all participating hospitals as per local assays.

Blood samples drawn for VDZ analyses were centrifuged and stored as serum at −20 °C. Serum samples were transported in batches on dry ice to be analyzed at the ISO 15189 certified laboratory, Sanquin Diagnostic Services (Amsterdam, The Netherlands).

Disease activity was assessed using the Harvey Bradshaw Index (HBI) for CD and the Simple Clinical Colitis Activity Index (SCCAI) for UC. Clinical remission was defined as HBI < 5, SCCAI < 3, and biochemical remission was defined as CRP < 5 mg/L and fecal calprotectin < 250 mg/kg [15,16,17].

Analyses of VDZ concentrations and anti-VDZ-antibodies were performed by Sanquin Diagnostic Services according to the cascade principle (antibody determination only when VDZ concentrations < 1 mg/L) using an in-house developed and validated ELISA and a drug-sensitive radioimmunoassay, which were previously described more in detail [13,18]. In order to reduce inter-test variability all VDZ trough concentration samples were run in one batch.

Maintenance VDZ trough concentrations were considered adequate if >14 mg/L for both CD and UC [9].

### 2.2. Statistics

Descriptive statistics were presented as mean with standard deviation (SD) for numerical normally distributed variables, median with interquartile range (IQR) for non-normally distributed variables, and as absolute numbers with percentages for categorical variables. Missing data were excluded from the relevant statistical analysis, when necessary. The sample size calculation is based on a dichotomous primary outcome parameter for disease remission, multiple logistic regression, an alpha of 0.05 (2-sided), and a power of 80%. In addition, the percentage of remission was described as a measure of the prevalence of remission, overall and stratified by UC and CD. The 95% confidence intervals (95% CIs) were calculated based on binomial distribution because of the dichotomous outcome variable.

Multiple logistic regression was applied to investigate the relationship between disease activity (active disease/remission) and VDZ trough concentrations and whether there were confounders. These analyses were stratified for the IBD category, as each type has its own disease activity scoring system. Scatterplots were created to visualize the relationship between trough concentrations and disease activity scores. Associations were analyzed with the Mann–Whitney test and the Chi-square test. All analyses were performed with SPSS statistics version 22 (IBM, Armonk, New York, NY, USA). Statistical significance was set at a *p*-value < 0.05.

### 2.3. Ethics

The protocol for this study was approved by the Medical Ethical Review Committee (MERC) of Máxima Medical Center according to the Dutch Act on Medical Research Involving Human Subjects (WMO) and subsequently approved by the MERCs of the other participating medical centers. All patients included gave written informed consent. Our study was funded with a grant from the Innovation and Research Committee (COI) of Máxima Medical Center.

## 3. Results

### 3.1. Patient Population

A total of 159 patients (59 CD and 100 UC) were included in this study. The characteristics of the included patients are described in Table 1. All patients received a fixed dose of 300 mg VDZ generally every eight weeks (range 4–12). The median disease duration was 13 years (IQR 7.0–21.0). The median vedolizumab treatment duration was 2.4 years (IQR 0.9–4.8). The majority of the patients (*n* = 131, 82.4%) were previously exposed to anti-TNF treatment. In 15 patients, a recent fecal calprotectin determination was missing; thus, these patients were only used for the calculation of clinical remission associations.

### 3.2. Vedolizumab Trough Concentrations

The VDZ concentrations during the maintenance phase ranged from 0.1 mg/L to 74.4 mg/L and are shown in Figure 1.

In the total group, the median VDZ concentration in patients who were in clinical remission was 16.5 mg/L (IQR 12.0–23.3); for patients not in remission, this was 16.7 mg/L (IQR 9.0–24.7) (*p* = 0.823). The median VDZ concentration in CD patients who were in clinical remission was comparable to those who were not (16.3 mg/L (IQR 11.2–22.1) and 19.0 mg/L (IQR 9.1–28.0), respectively (*p* = 0.587)). For UC patients, the median VDZ concentration of patients who were in clinical remission was also comparable to those who were not in clinical remission (16.6 mg/L (IQR 12.0–23.7) and 16.4 mg/L (IQR 8.7–21.7), respectively (*p* = 0.486)).

Patients in biochemical remission in the total group had a median VDZ concentration of 17.4 mg/L (IQR 13.0–25.1), whereas patients not in biochemical remission had a median VDZ concentration of 14.3 (IQR 8.8–23.6) (*p* = 0.021). When stratified by IBD type, statistically significant higher VDZ trough concentrations were only found in UC patients in biochemical remission (17.6 mg/L (IQR 13.2–24.8) vs. not in remission 13.7 mg/L (8.3–20.8) (*p* = 0.043)).

The proposed therapeutic VDZ threshold of >14 mg/L was achieved in 96 patients (60.4%). One patient had an extremely low drug concentration (0.1 mg/L); therefore, anti-VDZ antibodies were determined but were undetectable.

### 3.3. Clinical and Biochemical Remission and Relationship with Vedolizumab Trough Concentrations

Clinical remission was present in 103 patients (65%), whereas a smaller part of the patients was in biochemical remission (*n* = 67, 45%).

In the total IBD group, an association between VDZ trough concentration and biochemical remission was found (OR: 1.036, 95% CI 1.005–1.067) (Figure 2); when stratified by IBD type, this association was not found. However, no association was found between the VDZ trough concentration and clinical remission, either for the total group of patients (OR:1.004, 95% CI 0.978–1.032) (Figure 3) or stratified by IBD type (CD: OR 0.995 (95% CI 0.956–1.036, UC: OR 1.012 (95% CI 0.975–1.050).

Patients that had a trough concentration higher than the proposed therapeutic threshold were not more likely in remission (total group of patients with IBD OR 0.991 *p* = 0.605, stratified by disease CD: OR 0.983 *p* = 0.507, UC: OR 0.997 *p* = 0.916).

Only age appeared to be a confounder in the UC patient group since a lower rate of clinical remission was seen in older patients (*p* = 0.017). The other studied patient characteristics, albumin, weight, smoking status, disease duration, and anti-TNF use in history, were not associated with disease remission. The concomitant use of corticosteroids, aminosalicylates, and immunosuppressants was not associated with clinical disease remission either.

In the quartile analysis as shown in Figure 4, the highest percentage of patients in remission was in quartile 2. Clinical remission in patients with UC was achieved in 48%, 74.1%, 62.5%, and 66.7% of patients in quartiles 1 to 4 (*p* = 0.264), respectively, whereas for patients with CD, this remission rate was 60%, 86.7%, 73.3%, and 50% (*p* = 0.064), respectively (Figure 5).

## 4. Discussion

In this prospective, observational, cross-sectional, multicenter study, we explored the association between VDZ trough concentrations and both clinical and biochemical disease remission in a real-life IBD population on VDZ maintenance treatment. To our knowledge, this is one of the largest prospective observational studies in which this relationship has been explored.

In the present study, an exposure–response relationship was demonstrated between VDZ trough concentrations and biochemical remission. However, no association was found between VDZ trough concentrations and clinical remission. VDZ concentrations revealed large interindividual variability, with a relatively large proportion of patients (39.6%) having a VDZ trough concentration below the assumed therapeutic threshold of 14 mg/L [9,14].

There are several studies on VDZ in which an exposure–response relationship in the maintenance phase has been reported [9,14,19,20,21,22]. If present, an exposure–response relationship was mainly observed in the post-induction phase (week 14), at the beginning of the maintenance phase [23]. In this phase of treatment, allegedly, primary non-responders may over time experience more beneficial therapeutic effects, as a response to VDZ may take up to 6 months in some patients [24]. In addition, in these studies, outcome measure remission was defined differently than in the current study. Remission was defined as corticosteroid-free remission, histological response/remission, or the prediction of response after one year after exposure in induction [25]. In the present study, the percentage of patients in the post-induction phase was too small for sub-analysis, and when we studied our patient population that was treated with VDZ up to one year (N = 43, 27%), an association between VDZ trough concentration and clinical remission was not found.

In a similar cross-sectional study with a large cohort of patients with IBD on VDZ maintenance treatment, similar results were demonstrated: an exposure–response relationship for biochemical remission, but no (or at most weak) exposure–response relationship for clinical remission was observed [14,26]. However, in a smaller cross-sectional study, neither clinical, biochemical, nor endoscopic association with trough concentrations were found, probably due to the small number of patients included (N = 73) [26].

In various studies, it has been reported that patient-reported symptoms were poor predictors of mucosal inflammation [27,28]. These tools might be more valuable for investigating trends in disease activity rather than measuring clinical remission. Indeed, biochemical remission seems to be a better predictor of mucosal inflammation [29].

The therapeutic lower limit concentrations in the maintenance phase reported in the literature, mainly measured in weeks 14 and 52, were between 12–15 mg/L [9,14]. Likewise, in our population, the highest remission rate (both in CD and UC patients) was found in quartile 2 with levels between 10 and 16 mg/L. Higher VDZ concentrations were not associated with a higher chance of a beneficial therapeutic response as most likely a pharmacodynamic plateau is reached [26]. Consequently, we did not find higher remission percentages in quartiles 3 and 4 when compared with quartile 2 for patients with CD and UC.

Furthermore, there seems to be a discrepancy between concentrations required for receptor saturation (1 mg/L) and the previously proposed therapeutic lower limit of around 14 mg/L [5,25]. The complete pharmacodynamic mechanism of VDZ remains unclear. To understand this mechanism, in a small study with 37 patients, the correlation between tissue and serum VDZ concentration and their association with mucosal inflammation and response to VDZ were evaluated: patients with higher tissue exposure had better biochemical and endoscopic outcomes. Additionally, these tissue concentrations were higher than theoretically needed for receptor saturation [30].

No covariates in the predefined intrinsic and extrinsic patient characteristics were identified in this study that had a statistically significant influence on disease remission. In contrast to several studies, previous anti-TNF exposure was not associated with disease remission. Although it is suggested that the group of patients that failed prior anti-TNF therapy is more challenging to treat and they might need more exposure or longer time to respond to VDZ treatment than anti-TNF-naïve patients, it has also been shown that prior anti-TNF use does not affect the pharmacokinetics of VDZ [5]. Another explanation for not finding this association could be that patients who were primary non-responders to VDZ had already discontinued VDZ treatment and did not meet the inclusion criteria of this study. In similar studies, the absence of this association between anti-TNF-naïve patients and clinical remission was confirmed [5,11,14,20,22,26]. Furthermore, the group of patients that were anti-TNF naïve (17.6%) was small in our study.

A strength of this study is the large population we prospectively investigated. The population was included in four regional, non-academic expert centers for IBD, representing a real-life cohort. Trough concentrations were consistently measured in a central laboratory, applying one batch measurement, resulting in minimal laboratory variation by analysis variance.

Several flaws can also be identified regarding our study. In the maintenance phase, most patients had already been treated with VDZ for longer than a year, potentially thwarting a relationship between trough concentration and disease activity by excluding failures of VDZ treatment earlier on. Importantly, in all centers that participated in the study, reactive TDM is applied in VDZ treatment and dosing frequencies are adjusted accordingly to optimize the treatment. Quartile analysis of trough concentrations in relation to disease activity therefore may conceal the chance of beneficial effect. This might be the reason we did not find a clear therapeutic VDZ threshold in the present study.

Finally, pharmacokinetic differences will not necessarily explain differences in pharmacodynamic effect, especially when the majority of patients potentially receive more drugs than necessary.

## 5. Conclusions

The therapeutic drug monitoring of VDZ in patients with IBD on maintenance therapy revealed large interindividual differences in VDZ trough concentrations. In this study, an exposure–response relationship was observed between VDZ trough concentrations and biochemical remission but not for clinical remission. A therapeutic VDZ threshold concentration for remission could not be defined. The value of dose-optimizing strategies based on proactive VDZ trough concentration assessment seems therefore limited in clinical practice.

## Figures and Tables

**Figure 1 pharmaceutics-15-00972-f001:**
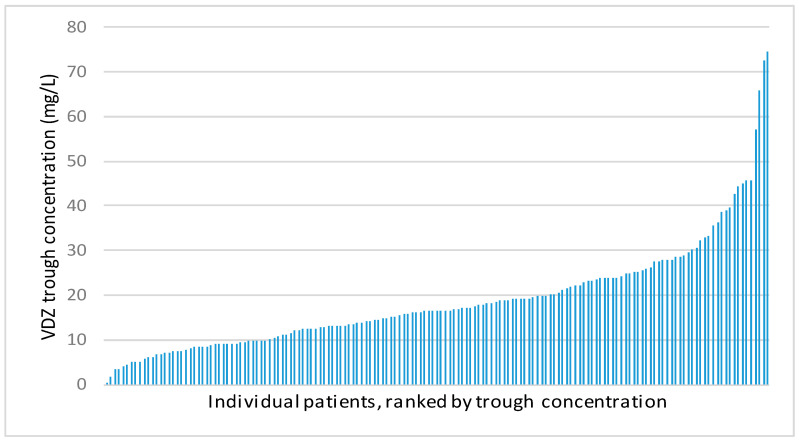
Distribution of vedolizumab trough concentrations in total cohort in mg/L.

**Figure 2 pharmaceutics-15-00972-f002:**
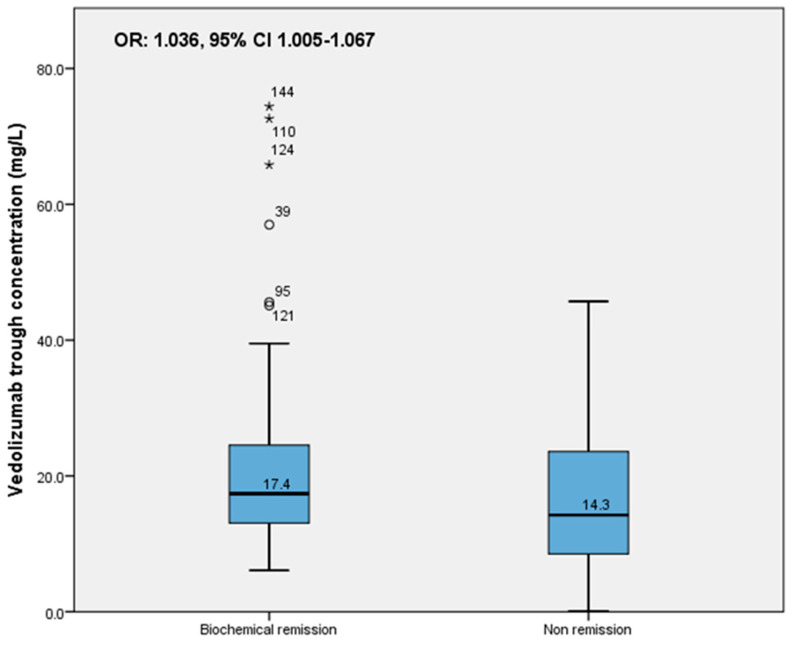
Association of trough concentration with biochemical remission in total IBD group. (° indicates an outlier in the dataset, * indicates there is an extreme outlier in the dataset).

**Figure 3 pharmaceutics-15-00972-f003:**
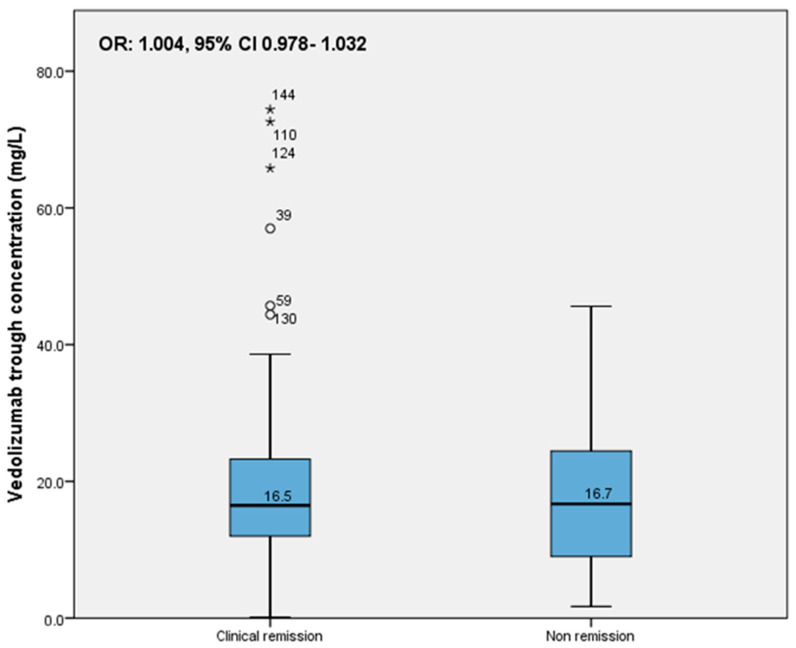
Association of trough concentration with clinical remission in total IBD group. (° indicates an outlier in the dataset, * indicates there is an extreme outlier in the dataset).

**Figure 4 pharmaceutics-15-00972-f004:**
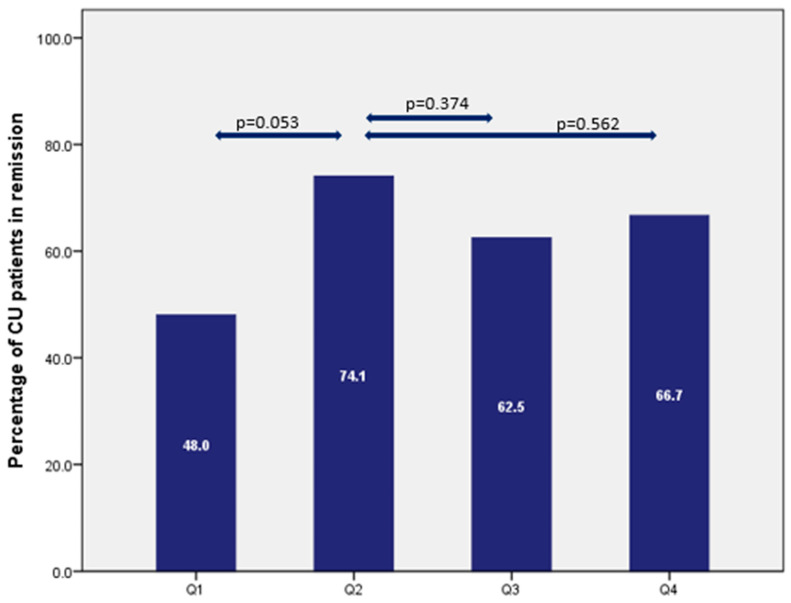
Percentage of UC patients in clinical remission expressed per quartile (Q1 < 10 mg/L, Q2 10.1–16.6 mg/L, Q3 16.7–23.3 mg/L, Q4 > 23.4 mg/L).

**Figure 5 pharmaceutics-15-00972-f005:**
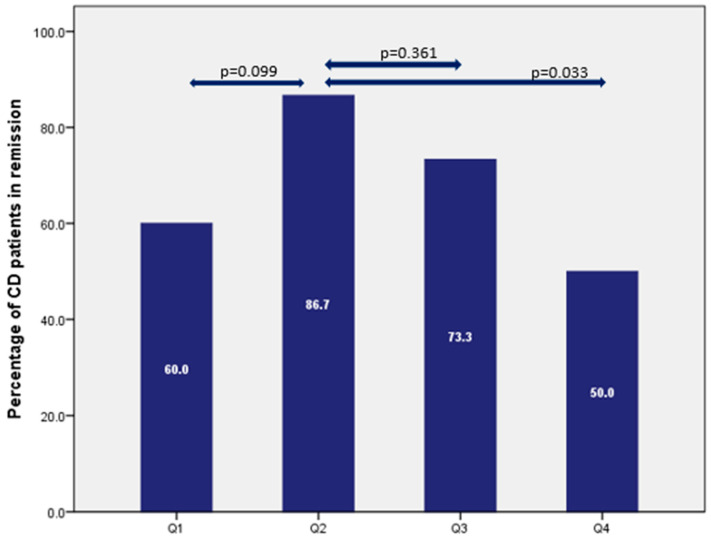
Percentage of CD patients in clinical remission expressed per quartile (Q1 < 10.5 mg/L, Q2 10.6–16.5 mg/L, Q3 16.6–24.7 mg/L, Q4 > 24.8 mg/L).

**Table 1 pharmaceutics-15-00972-t001:** Baseline patient characteristics.

	CD	UC	Total
**Number of patients, (*n* (%))**	59 (37.1)	100 (62.9)	159
**Age, mean (SD)**	48 (15.2)	53 (16.2)	51 (16.0)
**Sex (male, *n* (%))**	15 (25.4)	54 (54.0)	69 (43.4)
**BMI (kg/m^2^, median, [IQR])**	24.7 [21.4–29.4]	25.6 [23.5–28.3]	25.3 [23.1–28.6]
**Disease duration (year, median [IQR])**	14.0 [10.0–29.0]	11.8 [5.0–18.0]	13 [7.0–21.0]
**Vedolizumab treatment duration (year, median [IQR])**	2.6 [1.1–5.1]	2.3 [0.7–4.5]	2.4 [0.9–4.8]
**Interval vedolizumab (weeks, median, [range])**	8 [4–8]	8 [4–12]	8 [4–12]
**Active smoker (*n* (%))**	5 (8.5)	9 (9.0)	14 (8.8)
**Prior biological exposure (*n* (%))**			
- **Anti-TNF**	48 (81.4)	83 (83.0)	131 (82.4)
- **IL-6 inhibitor**	10 (16.9)	1 (1.0)	11 (6.9)
**Co-medication (*n* (%))**			
- **Immune suppressives**	4 (6.8)	16 (16.0)	20 (12.6)
- **Corticosteroids**	4 (6.8)	18 (18.0)	22 (13.8)
- **Aminosalicylates**	7 (11.9)	36 (36.0)	43 (27.0)
**Clinical remission (*n* (%))**	40 (67.8)	63 (63.0)	103 (64.8)
**Nonremission (*n* (%))**	19 (32.2)	37 (37.0)	56 (35.2)
**Biochemical remission (*n* (%))**	19 (33.9)	48 (51.6)	67 (45.0)
**Nonremission (*n* (%))**	37 (66.1)	45 (48.4)	82 (55.0)
**CRP (mg/L, median [IQR])**	6 [2.0–6.3]	2.9 [0.98–6.0]	4 [1.0–6.0]
**Hemoglobin (mmol/L, median [IQR])**	8 [8.0–9.0]	9 [8.0–9.0]	9 [8.0–9.0]
**Albumin (g/L, median [IQR])**	43 [40.0–44.0]	44 [42.0–46.0]	44 [41.0–45.0]
**Fecal calprotectin (mg/kg, median [IQR])**	58 [24.5–182.5]	63 [14.0–230.0]	59 [17.5–215.0]

SD standard deviation, IQR interquartile range, TNF tumor necrosis factor, IL interleukin, CRP C-reactive protein.

## Data Availability

Data are available on reasonable request from the corresponding author.

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
