# Peer review of "Therapeutic Drug Monitoring of Vedolizumab in Inflammatory Bowel Disease Patients during Maintenance Treatment—TUMMY Study"

_pharmaceutics, 2023, doi:10.3390/pharmaceutics15030972_

Round 1

Reviewer 1 Report

The manuscript describes the results of observational multicenter study that was designed to assess/confirm the association between vedolizumab trough concentrations and therapeutic effects in IBD patients receiving this drug. As many similar studies were performed in the past, the Authors should indicate the novelty of the proposed research. Moreover, several other issues should be clarified.  

It is not clear from the manuscript whether the intensity and frequency of adverse effects are dependent on vedolizumab concentrations.

Pharmacokinetics of vedolizumab may be also described in the Introduction section or the Discussion.

Materials and Methods section

Vedolizumab dose used was not shown despite the fact that “VDZ dose (mg) …. were obtained from the electronic patient records”.

The time from the last infusion to the time of sampling was not indicated.

Why was necessary to take so much blood (5 ml)?

It is not clear whether whole blood, serum or plasma was used for drug determination and whether the samples were frozen or processed immediately.

The methods used for the measurement of biochemical markers should be listed.

Some details about the ELISA test used (manufacturer, sensitivity, calibration curve range, sample volume) should be added.

If dosing regimens were adjusted, the Authors should present doses  before and after the intervention and show how this influenced the trough concentrations.

Reviewer 2 Report

The authors tried to to determine whether there is an association between VDZ trough concentration and clinical and biochemical remission in the maintenance phase. It is an actual topis. The study is well designed and results are well presented, although are not promising considering relationship between TDM of vedolizumab and clinical remission. I have a few comments regarding this paper:

Introduction

1. Some sentences in introduction should be written clearer.

“Exposure-response relationships can exist when a higher drug concentration results in more pharmacodynamic effect or when a higher disease activity results in higher drug clearance.”

2. Tertiary aim could be contained in the first one. Therefore, it was not necessary.

Methods

3. “Blood samples were drawn just before the administration of a regular VDZ infusion at the outpatient clinic on the day of treatment. “ Is this Ctrough? If it is, you should write.

4. “Sanquin Diagnostic Services.” One more sentence regarding this would be appropriate.

5. “..amount with percentage for categorical variables” You should replace amount with number or absolute number.

Reviewer 3 Report

The authors investigated whether there was an association between vedolizumab trough concentration and clinical and biochemical remission in the maintenance phase. The author did not explain clearly the significance of this study, and the paper was not complete. Therefore, the article can be published after a major revision.

The specific content is as follows:

1.      In the introduction part, the authors should explain the significance of this study.

2.      The paragraph summary format of the article should be numbered according to the journal requirements.

3.      The authors should explain some information about the patient, including gender, age, whether these have an impact on the research should be explained.

4.      The author mentions the use of a validated ELISA and a drug-sensitive radioimmunoassay. This part of the experimental operation needs to be described in detail, including which reagents and consumables are used. The "2.Materials and Methods" section should describe the detection method in detail.

Round 2

Reviewer 1 Report

The Authors properly addressed most of my comments. The sentence "Exposure-response relationships for drugs can exist when a higher drug concentration results in more (pharmacodynamics) drug effect or when a higher disease activity results in higher drug clearance and thus a lower drug concentration" is unclear. The Authors should provide examples of drugs for which higher disease activity results in a higher drug clearance. In some cases the opposite effects are observed, e.g. when liver or kidney function is decreased as a results of disease progression.

Author Response

Dear reviewer,

Thank you for your time to review our manuscript. We are happy that we could address most of the comments from round one. 

Point 1: The sentence "Exposure-response relationships for drugs can exist when a higher drug concentration results in more (pharmacodynamics) drug effect or when a higher disease activity results in higher drug clearance and thus a lower drug concentration" is unclear. The Authors should provide examples of drugs for which higher disease activity results in a higher drug clearance. In some cases the opposite effects are observed, e.g. when liver or kidney function is decreased as a results of disease progression.

Response 1: 

We have added an example in that sentence: “Exposure-response relationships for drugs can exist when a higher drug concentration results in more (pharmacodynamics) drug effect or when a higher disease activity results in higher drug clearance and thus a lower drug concentration (e.g. in infliximab).

For monoclonal antibodies renal and hepatic clearance plays a minimum to no role. We would like to refer to a review article of one of the co-authors dr. Derijks which is reference 4 in our manuscript (PMID: 29512050). The following paragraph is taken from that article that gives a more elaborate explanation of the clearance of TNF-alfa clearance:

"In addition, target-mediated clearance may contribute substantially to MAb clearance. This clearance pathway, which is often referred to as the ‘antigen sink’, is associ[1]ated with disease severity [293]. Thus, patients with more extensive IBD, who often have a higher TNFa burden, will tend to have a higher fraction of MAb clearance through target-mediated clearance and are prone to suboptimal response. These patients may need higher doses of TNFa inhibitors to neutralize excess amounts of TNFa in both tissue and serum, at least for that moment."

We hope that this clarifies the sentence more.

Kind regards,

Merve Sivridas, on behalf of all authors

Reviewer 3 Report

Compared with the previous version, the revised article has been greatly improved. Therefore, I agree to accept it in the present form.

Author Response

Dear reviewer,

Thank you for your time to review our manuscript. We are very happy to publish in pharmaceutics. 

Kind regards,

Merve Sivridas, on behalf of all authors